# Population-Based Assessment of Determining Predictors for Discharge Disposition in Patients with Bladder Cancer Undergoing Radical Cystectomy

**DOI:** 10.3390/cancers14194613

**Published:** 2022-09-23

**Authors:** Raj A. Kumar, Kian Asanad, Gus Miranda, Jie Cai, Hooman Djaladat, Saum Ghodoussipour, Mihir M. Desai, Inderbir S. Gill, Giovanni E. Cacciamani

**Affiliations:** 1Catherine & Joseph Aresty Department of Urology, Keck Medicine of USC, University of Southern California, Los Angeles, CA 90033, USA; 2Bladder and Urothelial Cancer Program, Rutgers Cancer Institute of New Jersey, New Brunswick, NJ 08903, USA

**Keywords:** radical cystectomy, discharge disposition, skilled nursing home, home discharge, marital status, disparities, insurance, population-based cohort study, Premier Healthcare Database

## Abstract

**Simple Summary:**

Our study analyzed 138,151 radical cystectomy patient encounters to determine which patient and facility characteristics are associated with discharge home and discharge to continued rehabilitation facilities. We used multivariate logistic regression to statistically analyze these datapoints while controlling for other variables. We found that older age, single/widowed marital status, female gender, increased Charlson Comorbidity Index, Medicaid, and Medicare insurance and open surgery are associated with Continued Rehabilitation Facility (CRF) discharge.

**Abstract:**

Objective: To assess predictors of discharge disposition—either home or to a CRF—after undergoing RC for bladder cancer in the United States. Methods: In this retrospective, cohort study, patients were divided into two cohorts: those discharged home and those discharged to CRF. We examined patient, surgical, and hospital characteristics. Multivariable logistic regression models were used to control for selected variables. All statistical tests were two-sided. Patients were derived from the Premier Healthcare Database. International classification of disease (ICD)-9 (<2014), ICD-10 (≥2015), and Current Procedural Terminology (CPT) codes were used to identify patient diagnoses and encounters. The population consisted of 138,151 patients who underwent RC for bladder cancer between 1 January 2000 and 31 December 2019. Results: Of 138,151 patients, 24,922 (18.0%) were admitted to CRFs. Multivariate analysis revealed that older age, single/widowed marital status, female gender, increased Charlson Comorbidity Index, Medicaid, and Medicare insurance are associated with CRF discharge. Rural hospital location, self-pay status, increased annual surgeon case, and robotic surgical approach are associated with home discharge. Conclusions: Several specific patient, surgical, and facility characteristics were identified that may significantly impact discharge disposition after RC for bladder cancer.

## 1. Introduction

Radical cystectomy (RC) is the mainstay of treatment for muscle-invasive bladder cancer and refractory non-muscle-invasive bladder cancer [1,2]. Despite significant refinement and standardization over the last several years, RC remains a morbid procedure with significant post-operative complications, readmissions, and mortality [3,4]. For this reason, careful patient selection is critical for successful surgical outcomes. There is little data regarding discharge disposition following RC—either to a Continued Rehabilitation Facility (CRF) (including skilled nursing facility (SNF), acute care rehabilitation, outpatient care rehabilitation, etc.) or directly home. A recent study linked discharge to SNFs with increased rate of readmission [5]. Discharge to SNF among other facilities was associated with nearly 50% higher odds of readmission at both 30 and 90 days following discharge after RC.

The cost of CRF services vary widely by state, length of stay, labor, services required, and vacancy. Per the U.S. Census Bureau, 9.2% of the U.S. population remained uninsured in 2019 [6]. Additionally, many insurance plans deter admission to CRFs, and so patients can incur significant financial burden [7]. Given the likelihood of readmission and financial strain, it is imperative to assess which patients are at risk of admission to CRFs. While several studies have associated measures such as frailty, increased age, and poor exercise tolerance with CRF discharge, there has not been a robust, systematic analysis of predictors of discharge disposition [8,9,10].

We hypothesized that older patients or patients without a system of care (such as having a widowed status) would be more likely to be discharged to an SNF. Additionally, we hypothesized that a self-pay insurance status would indicate CRF discharge. We also felt that the surgical approach may prove to have an impact in terms of recovery and therefore minimize CRF discharge [11].

In this study, we used a population-based approach to evaluate predictors of discharge disposition following RC.

## 2. Materials and Methods

### 2.1. Data Source

We used Premier Healthcare Database (PHD) by Premier Inc. (Charlotte, NC, USA), a large U.S. (n. 1041 contributing hospitals/healthcare systems), hospital-based, service-level, all-payer database that includes inpatient discharge information. Inpatient admissions include over 121 million visits, representing approximately 25% of all annual U.S. admissions [12]. PHD collects a large volume of data that could be identified and analyzed using ICD 9 and 10 codes as has been done in multiple past studies [13].

### 2.2. Study Cohort and Variables

We identified patients diagnosed with bladder cancer (BCa) between 2000 and 2019 and underwent RC. We excluded patients who died during the hospital stay. Patients were allocated into two groups based on discharge disposition after RC: those discharged home (with/without home health services) or to CRFs (Appendix A). We used medical-record-level details of International Classification of Diseases, 9th and 10th (ICD-9 and ICD-10) and diagnosis and Current Procedural Terminology (CPT) codes to identify patients (aged ≥18 years) undergoing RC for BCa, urinary diversion (continent or incontinent) and surgical approach (open or robotic [14,15]) (Appendix B). Data on patient characteristics (age, gender, race, and ethnicity, Charlson comorbidity Index (CCI), marital status, primary health insurance) surgical characteristics (urinary diversion, surgical approach) and facility characteristics (hospital size, annual hospital volume and surgeon volume, hospital location and teaching status, year of surgery, and region (Midwest, Northeast, South, West)) were analyzed.

### 2.3. Statistical Analysis

Annual hospital and surgeon RC volumes were calculated and presented as quintiles. Volumes at or below the 20th percentile were considered “low” and volumes above the 80th percentile were considered “high”. Values between these extremes were considered “intermediate”, as previously described [16]. Continuous and categorical variables were presented as mean and standard deviation, and median and interquartile range (IQR), respectively. A univariate analysis was performed to compare differences in baseline demographics, surgical factors, and facility characteristics between the two cohorts. In the univariate analysis, Kruskal–Wallis, chi-squared (X^2^), and Fisher’s exact tests were used to compare continuous and categorical variables as appropriate. We performed separate multivariable logistic regression models. The multivariable model included variables previously found to be predictors of discharge to CRFs [8,9,10] and significant variables from our preliminary univariate analysis. Nationally representative estimates were achieved using projection weights linked to the Premier Database derived from the 1998 American Hospital Association Annual Survey and validated by the 1998 National Hospital Discharge Survey as previously described [17,18]. A two-tailed test with *p* < 0.05 was considered statistically significant. Data were analyzed using SAS 9.0 software and reported according to guidelines for reporting statistics for clinical research in urology [19].

## 3. Results

### 3.1. Baseline Characteristics

We identified 138,151 patients diagnosed with bladder cancer (BCa) between 2000 and 2019 and underwent RC. Baseline characteristics of the patient population are reported in Table 1. Facility characteristics are reported in Table 2. A weighted total of 138,151 patients was included. A total of 24,922 (18.0%) patients were admitted to SNFs. Median age was 70.0 (IQR:62.0–76.0). Median length of stay was 8.0 days (IQR:7.0–12.0).

Of those discharged home, 94,250 (83.2%) were male and 18,976 (16.8%) female; 65,539 (61.4%) were married and 32,701 (28.9%) single. 22,355 (19.7%) underwent robotic RC and 90,874 (80.3%) underwent open RC. Of those discharged to CRFs, 18,406 (73.9%) were male and 6516 (26.1%) female; 10,962 (44.0%) were married and 11,520 (46.2%) single; 4477 (18.0%) underwent robotic RC and 20,445 (82.0%) underwent open RC. Trends over time showed increasing annual percent of patients discharged to CRF, from 16% before 2005 to 18% in 2019, with a peak of 21.5% in 2017 (Figure 1).

### 3.2. Predictor of Discharge Disposition after RC

Multivariate analysis (Figure 2) revealed that older age, single marital status, female gender, increased CCI score, Medicaid insurance, Medicare insurance, non-teaching hospital status, and northeast geographic location are associated with a significant increase in discharge to CRFs.

Rural hospital location, self-pay status, continent neobladder diversion, 200–299 bed hospital size, increased annual surgeon case volume, and robotic surgical approach are associated with discharge home.

### 3.3. Surgical Volumes-Based Analysis

Multivariate analysis was performed after separating data into high-volume (HV) and non-high-volume (NHV) cohorts (Appendix C). In the HV cohort, increased age, single marital status, female gender, CCI ≥ 2, Medicaid insurance, Medicare insurance, non-teaching hospital status, and northeast geographic location are associated with a statistically significant increase in CRF discharge.

“Other” marital status [rural hospital location, self-pay status, south geographic location, west geographic location, and robotic approach] are associated with discharge home.

In the NHV cohort (Appendix C), increased age, single marital status, “other” marital status, female gender, increased CCI score, “other” race, Medicaid insurance, Medicare insurance, non-teaching hospital status, and northeast geographic location are associated with a statistically significant increase in CRF discharge.

Rural hospital location, self-pay status, 200–299 bed hospital size [south geographic location, west geographic location, and robotic surgical approach] are associated with discharge home.

### 3.4. Geographic Area Analysis

Multivariate analysis was performed by geographic region: Midwest, Northeast, South, and West. Odds ratios, confidence intervals, and statistical significance are reported in Figure 3. Below we have reported our statistically significant findings.

#### 3.4.1. Midwest

In the Midwest region, increased age, single marital status, female gender, CCI score of 2, “other” insurance, “high” annual surgeon volume, and later year of surgery are associated with a statistically significant increase in CRF discharge.

“Other” race, 400 or more bed hospital size, rural hospital location, “intermediate” annual surgeon volume, and robotic approach are associated with discharge home.

#### 3.4.2. Northeast

In the Northeast region, increased age, single marital status, “other” marital status, female gender, CCI score of 2, Medicaid insurance, Medicare insurance, “intermediate” annual hospital volume, and later year of surgery are associated with a statistically significant increase in CRF discharge.

Self-pay status, 200–299 bed hospital size, 300–399 bed hospital size, non-teaching hospital status, rural hospital location, ‘high” annual hospital volume, “high” annual surgeon volume, “intermediate” annual surgeon volume, and robotic approach are associated with discharge home.

#### 3.4.3. South

In the South region, increased age, single marital status, “other” marital status, female gender, CCI score of 2, Medicaid insurance, Medicare insurance, “other” insurance, non-teaching hospital status, and later year of surgery are associated with a statistically significant increase in CRF discharge.

Self-pay status, 300–399 bed hospital size, “intermediate” annual hospital volume, “high” annual surgeon volume, “intermediate” annual surgeon volume, and robotic surgical approach are associated with discharge home.

#### 3.4.4. West

In the West region, increased age, single marital status, female gender, CCI score of 2, Medicaid insurance, Medicare insurance, and non-teaching hospital status are associated with a statistically significant increase in CRF discharge.

CCI score of 1, continent neobladder diversion, 200–299 bed hospital size, “high” annual hospital volume, and “high” annual surgeon volume are associated with discharge home.

## 4. Discussion

This study evaluates the impact of patient, surgical, and facility factors on discharge disposition of patients with BCa undergoing RC and urinary diversion in the U.S.

Our study has several important findings. First, females, single or widowed patients, and those with higher CCI were significantly more likely to be discharged to a SNF. For these patients, pre-operative counselling should include discussions regarding the increased likelihood of discharge to CRFs. In 2011, Aghazadeh et al. found that older age, poor preoperative exercise tolerance, and longer hospital stay predicted CRF discharge [8]. However, that study included only 445 patients from the same institution (2004–2007). Several studies focused on frailty as an important predictor of discharge to CRFs [9,10]. Though not directly associated with CRF discharge, increased age and female gender were associated with increased frailty. This indirectly supports our findings that age and gender were associated with SNF discharge.

We found patients’ insurance status to be a significant predictor of discharge disposition. Patients who were self-pay were significantly less likely to be discharged to CRFs, while patients with Medicare and/or Medicaid were more likely to be discharged to CRFs. Both Medicaid and Medicare cover SNF stay up to a certain point [20,21]. Medicare Part A covers the entire cost of the first 20 days, and patients will be responsible for a $185.50 co-pay for the next 80 days. Patients will be entirely responsible for any subsequent SNF costs beyond the first 100 days. In 2018, one-fifth of hospitalized Medicare beneficiaries were discharged to SNFs, and Medicare paid a total of $28.5 billion on SNF services [22]. Self-pay patients are responsible for the entire cost and are therefore less likely to desire CRF stay.

Our analysis showed that higher volume surgeons and teaching hospitals were less likely to discharge patients to a CRF. This may be attributable to improved skill, reduced complication rates, use of standardized discharge pathways, and the implementation of standardized protocols including enhanced recovery after surgery (ERAS) protocols [23]. ERAS guidelines for RC were introduced in 2013 [24]. ERAS protocols have been shown to reduce length of stay for patients undergoing radical cystectomy without significant difference in complication rates and readmission rates [25,26].

That said, there is heterogeneity in the application of ERAS protocols between institutions, and even within the same institution [25]. One of the limitations of this study was that PHD did not allow us to control for institutions that have adopted ERAS protocols.

Our study reports a slight increase in discharge to CRF over time. While several novel procedures and technologies have contributed to more optimal surgical outcomes such as minimally invasive surgery and ERAS protocols, we feel that the increase in CRF discharge complements this appropriately. Discharge to CRF provides patients with continued skilled care while also permitting room turnover for more patients. Additionally, with improved surgical outcomes, more patients are eligible for RC. This broader patient pool includes more elderly patients and those with comorbidities that require skilled care even following discharge.

The multivariable analysis has shown that patients undergoing radical cystectomy with a continent urinary diversion are less likely to be discharged to CFR (OR = 0.82, 95% CI 0.76–0.90). This may be explained by targeted patient selection. Continent urinary diversion often warrants a robust selection of candidates that can more strongly tolerate surgical intervention efficiently benefit from a continent urinary diversion. In combination, home discharge, good tolerance of surgery, improved outcomes, and continent diversion may all affect the quality of life in these patients [27,28].

Our study reported that patients who underwent a robotic approach to their surgery were significantly less likely to be discharged to a skilled nursing facility. This may be because robotic RC is associated with decreased length of hospital stay compared to open RC and report fewer complications compared to open surgery [29]. While robotic RC is known to have an increased operative time and cost to the patient, our study shows that this may lead to decreased future costs by avoiding CRF stay. Generally speaking, open RC has been shown to be more cost effective than robotic RC [30]. However, the decreased need for extended CRF stay may impact the cost of robotic RC. Further reports should account for this aspect in the cost-analysis.

Interestingly, we found that both rural hospitals and large, high-volume centers and both associated with home discharge. Though seemingly contradictory, we would reconcile this finding by noting that CRFs in rural areas are smaller and not always available [31]. Larger urban areas are more likely to have available and skilled CRFs, however would more likely utilize this option if patient recovery is slow or if there have been post-surgical complications that warrant skilled nursing staff.

Significant geographic differences were found in CRF discharge across the United States. Patients with a CCI of 2 or greater were nearly twice as likely to be discharged to a CRF in the Northeast compared to the West (OR 3.085 and 1.622, respectively). High annual surgeon volume had more than a twofold greater increase in CRF discharge prediction in the Midwest compared to the West (OR of 1.103 and 0.485, respectively). Additionally, the South has seen the greatest annual increase in CRF discharge over time (OR 1.068 per year), while the West has seen the lowest increase over time (OR 1.016 per year). Insurance status showed the highest degree of variability across geographic region, and protocols based on insurance are highly variable per state and regional regulations.

Finally, we must also recognize the changes to the CRF system over time. Most significantly, in 2006, within 30 days of admission to a nursing facility, nearly 24% of short-stay patients were readmitted to a hospital. Following this, outpatient emergency department use and rehospitalization were added as quality measures for CRFs [32].

Although our findings impact patient preoperative counseling, costs, and outcomes, this must be interpreted within the study limitations. First, the PHD does not provide data on the granularity of cancer staging. Second, we do not have information about institutional adoption and enforcement of ERAS protocols. Third, we do not have data regarding in-hospital complications that could have an impact on discharge disposition as has been previously described [11]. Finally, our study is a retrospective analysis.

There are however several strengths to our study. The large study population allows us to better identify statistically significant findings that would have been otherwise missed in a smaller sample. To our knowledge this is the first population-based study that assesses the impact of patients, surgical, and facility characteristics on discharge disposition after RC for BCa. This is the first study with this size that assesses factors associated with readmission-providing opportunity to mitigate this in both care provider and administrative level. Additionally, the use of a multivariate analysis allows us to control for several variables that may have otherwise been confounding factors.

## 5. Conclusions

Several specific patient, surgical, and facility characteristics were identified that may significantly impact discharge disposition after RC for bladder cancer. This new information should help guide surgeons and patients with preoperative counseling and shared decision-making process. Prompt identification of patients at risk for non-home discharge can be useful for implementing comprehensive discharge planning protocols that may help with more appropriate and efficient resource allocation.

## Figures and Tables

**Figure 1 cancers-14-04613-f001:**
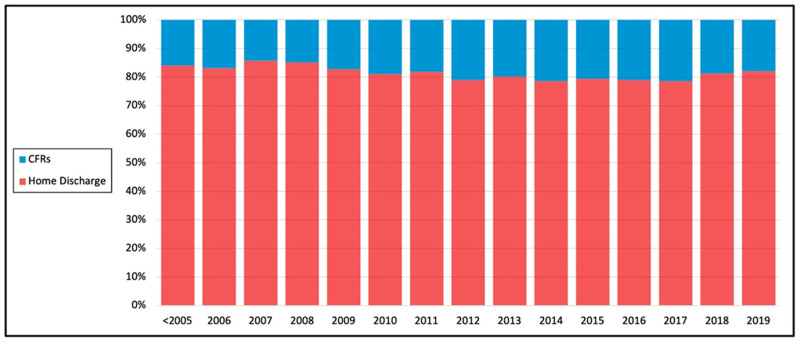
Trends over time of patients discharged home vs. to CFRs in the United States.

**Figure 2 cancers-14-04613-f002:**
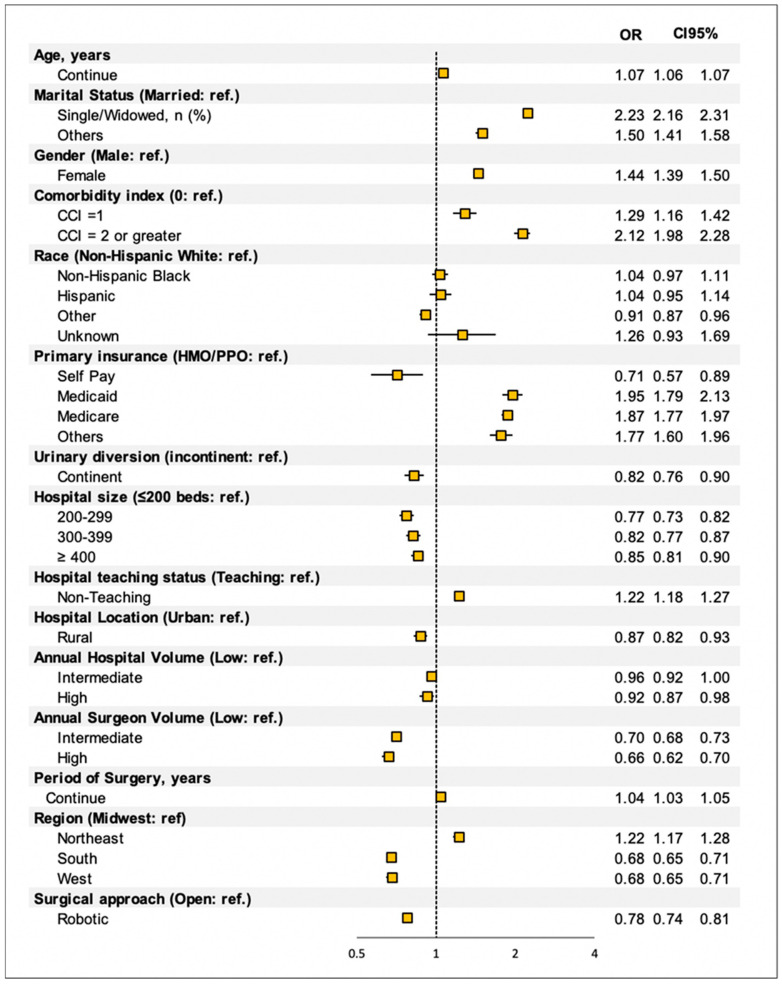
Multivariable Analysis Overall population. OR: Odds Ratio.

**Figure 3 cancers-14-04613-f003:**
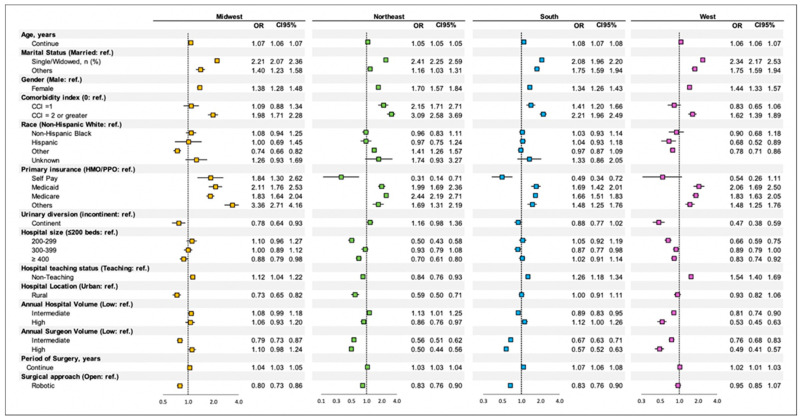
Multivariable Analysis—Geographical Subsets. OR: Odds Ratio.

**Table 1 cancers-14-04613-t001:** Patients and Surgical Characteristics.

	Home Discharge		CFRs		*p* Value
**N. of Patients**	**113,229**	**(82.0%)**	**24,922**	**(18.0%)**	
**Age, years, n (%)**					<0.0001
younger than 55	12,711	(93.5%)	884	(6.5%)	
55–64	27,594	(91.6%)	2545	(8.4%)	
65–69	21,150	(87.0%)	3161	(13.0%)	
70–74	21,854	(81.0%)	5111	(19.0%)	
75 or Older	29,920	(69.4%)	13,221	(30.6%)	
**Gender, n (%)**					<0.0001
Male	94,250	(83.7%)	18,406	(16.3%)	
Female	18,976	(74.4%)	6516	(25.6%)	
**Comorbidity index, n (%)**					<0.0001
CCI = 0	9371	(89.5%)	1105	(10.5%)	
CCI = 1	5160	(84.6%)	937	(15.4%)	
CCI 2 or greater	98,698	(81.2%)	22,880	(18.8%)	
**Race and Etnicity, n (%)**					<0.0001
N-H-White	88,224	(81.6%)	19,836	(18.4%)	
N-H-Black	5780	(81.2%)	1339	(18.8%)	
Hispanic	3089	(82.6%)	651	(17.4%)	
Other	15,902	(84.0%)	3028	(16.0%)	
Unknown	234	(77.5%)	68	(22.5%)	
**Primary insurance, n (%)**					<0.0001
Self-Pay	1958	(95.9%)	83	(4.1%)	
Medicaid	5658	(86.3%)	902	(13.8%)	
Medicare	68,890	(76.7%)	20,907	(23.3%)	
HMO/PPO	32,706	(93.1%)	2424	(6.9%)	
Others	4017	(86.9%)	606	(13.1%)	
**Marital Status**					<0.0001
Married, n (%)	69,539	(86.4%)	10,962	(13.6%)	
Single/Widowed, n (%)	32,701	(73.9%)	11,520	(26.1%)	
Others	10,989	(81.8%)	2440	(18.2%)	
**Surgical Approach**					<0.0001
Robotic, n (%)	22,355	(83.3%)	4477	(16.7%)	
Open, n (%)	90,874	(81.6%)	20,445	(18.4%)	
**Type of Urinary Diversion**					<0.0001
Incontinent n (%)	101,007	(81.4%)	23,113	(18.6%)	
Continent n (%)	6905	(90.9%)	695	(9.1%)	
**LOS, days, mean (SD)/ median (IQR)**	9.7 (6.3)	8.0 (6.0–11.0)	17.2 (13.9)	13.0 (8.0–21.0)	<0.0001
**LOS ≤ 5days n, (%)**	14,501	(95.2%)	733	(4.8%)	<0.0001
**LOS > 5days n, (%)**	98,728	(80.3%)	24,189	(19.7%)	<0.0001

**Table 2 cancers-14-04613-t002:** Facility Characteristics.

	Home Discharge		CFRs		*p* Value
**Hospital volume facility, beds, n (%)**					<0.0001
≤200	10,344	(78.6%)	2824	(21.4%)	
200–299	14,711	(82.4%)	3132	(17.6%)	
300–399	19,171	(82.2%)	4152	(17.8%)	
≥400	69,003	(82.3%)	14,814	(17.7%)	
**Hospital teaching status n (%)**					<0.0001
Teaching	63,303	(82.3%)	13,572	(17.7%)	
Non-teaching	49,926	(81.5%)	11,350	(18.5%)	
**Hospital Location n (%)**					0.1078
Urban	104,272	(81.9%)	23,026	(18.1%)	
Rural	8957	(82.5%)	1896	(17.5%)	
Not reported					
**Annual Hospital Volume n (%)**					<0.0001
High	23,230	(82.4%)	4951	(17.6%)	
Intermediate	67,882	(82.7%)	14,204	(17.3%)	
Low	22,117	(79.3%)	5767	(20.7%)	
**Annual Surgeon Volume n (%)**					<0.0001
High	22,354	(83.9%)	4285	(16.1%)	
Intermediate	69,758	(83.0%)	14,244	(17.0%)	
Low	21,117	(76.8%)	6393	(23.2%)	
**Year of Surgery n (%)**					<0.0001
<2005	32,569	(84.0%)	6187	(16.0%)	
2006	5692	(83.1%)	1161	(16.9%)	
2007	5920	(85.7%)	991	(14.3%)	
2008	6044	(85.0%)	1063	(15.0%)	
2009	6023	(82.7%)	1261	(17.3%)	
2010	5745	(81.1%)	1342	(18.9%)	
2011	5481	(81.9%)	1215	(18.1%)	
2012	5118	(78.9%)	1368	(21.1%)	
2013	5139	(80.1%)	1275	(19.9%)	
2014	5482	(78.5%)	1500	(21.5%)	
2015	5991	(79.3%)	1562	(20.7%)	
2016	6657	(79.0%)	1772	(21.0%)	
2017	6324	(78.5%)	1731	(21.5%)	
2018	6226	(81.2%)	1437	(18.8%)	
2019	4818	(82.0%)	1057	(18.0%)	
**Region n (%)**					<0.0001
Midwest	25,572	(79.0%)	6801	(21.0%)	
Northeast	21,484	(78.6%)	5833	(21.4%)	
South	42,969	(84.5%)	7862	(15.5%)	
West	23,204	(84.0%)	4426	(16.0%)	

## Data Availability

Restrictions apply to the availability of these data. Data were obtained from Premier, Inc., and are available from the authors with the permission of Premier, Inc.

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
