# Peer review of "Population-Based Assessment of Determining Predictors for Discharge Disposition in Patients with Bladder Cancer Undergoing Radical Cystectomy"

_cancers, 2022, doi:10.3390/cancers14194613_

Round 1
Reviewer 1 Report
Brilliant paper. I know this team and the work they do. It is sound.
I liked it very much in the way it contributed to the field. We are always looking for papers and ways in which the field can be advanced.
Accept for publication.
Author Response
We thank the esteemed reviewer for the comments.
Reviewer 2 Report
Thank you for giving me the possibility to review this interesting paper about the factors predicting discharge disposition in patients with bladder cancer undergoing radical cystectomy.
It represents a well written paper analyzing factors which may really have an impact on preoperative patient counseling.
I think the topic has really been well-analyzed with no gap. I found very interesting the fact robotic RC is associated with a less likely to be discharge in a SNF. It would be interesting to find out if from a matching between type of surgery and kind of discharge it would result any prediction of readmission.
I really think this paper deserves to be published.
Author Response
We thank the esteemed reviewer for the comments. We have addressed the minor typographical errors in the manuscript as requested.
Reviewer 3 Report
Authors aimed to assess predictors of discharge disposition – either home or to a continued rehabilitation facility (CRF) – after undergoing RC for bladder cancer in the United States. Multivariate analysis revealed that older age, single/widowed marital status, female gender, increased Charlson Comorbidity Index (CCI), Medicaid, and Medicare insurance are associated with CRF discharge. Rural hospital location, self-pay status, increased annual surgeon case, and robotic surgical approach are associated with home discharge. The issue addressed is original and of clinical interest. The methodology is correct and manuscript is well written. However, I suggest some minor revisions:
A. Figure 1 and 2: legends seem to be inappropriate.
B. The issue of urinary diversion should be considered at least in the discussion section (see for example: Creta M. Short- and Long-Term Evaluation of Renal Function after Radical Cystectomy and Cutaneous Ureterostomy in High-Risk Patients. J Clin Med. 2020 Jul 11;9(7):2191. doi: 10.3390/jcm9072191. PMID: 32664517; PMCID: PMC7408808. ---- Longo N, et al. Complications and quality of life in elderly patients with several comorbidities undergoing cutaneous ureterostomy with single stoma or ileal conduit after radical cystectomy. BJU Int. 2016 Oct;118(4):521-6. doi: 10.1111/bju.13462. Epub 2016 Apr 4. PMID: 26935245. )
Author Response
We thank the esteemed reviewer for the comments. We have now edited the figure legends as recommended. Additionally, we have considered your comments, included your suggested references, and added a paragraph in the discussion section stating the following:
"Our multivariable analysis has shown that patients undergoing radical cystectomy with a continent urinary diversion are less likely to be discharged to CFR (OR = 0.82, 95% CI 0.76-0.90). This may be explained by targeted patient selection. Continent urinary diversion often warrants a robust selection of candidates that can more strongly tolerate surgical intervention efficiently benefit from a continent urinary diversion. In combination, home discharge, good tolerance of surgery, improved outcomes, and continent diversion may all affect the quality of life in these patients."
Reviewer 4 Report
This study was reported the utility of the continued rehabilitation facility in patients with BCa who underwent radical cystectomy. Generally, this paper is well written. The reviewer thinks that this paper has useful information for readers. However, the reviewer would like to suggest some critiques to make this paper as follows.
Minor revision
1. On line 15and 32, the authors should delete “CCI.”
2. On line 16, the authors should spell out about “CRF.”
Author Response
We thank the esteemed reviewer for the comments. The requested changes have been made in the appropriate places.